# Clinical Impact of Implementing a Nurse-Led Adverse Drug Reaction Profile in Older Adults Prescribed Multiple Medicines in UK Primary Care: A Study Protocol for a Cluster-Randomised Controlled Trial

**DOI:** 10.3390/pharmacy10030052

**Published:** 2022-04-28

**Authors:** Vera Logan, Alexander Bamsey, Neil Carter, David Hughes, Adam Turner, Sue Jordan

**Affiliations:** 1Faculty of Medicine, Health and Life Sciences, Swansea University, Singleton Park, Swansea SA2 8PP, UK; alex.bamsey@wales.nhs.uk (A.B.); n.carter@swansea.ac.uk (N.C.); d.hughes@swansea.ac.uk (D.H.); a.j.turner@swansea.ac.uk (A.T.); 2Mount Surgery, Swansea Bay University Health Board, Port Talbot SA13 2BN, UK

**Keywords:** drug-related side effects and adverse reactions, polypharmacy, primary health care, aging, nurses, randomised controlled trial

## Abstract

(1) Aims: Adverse drug reactions (ADRs) particularly affect older people prescribed multiple medicines. The professional bodies of nursing, medicine and pharmacy have issued guidelines on identification and management of ADRs; however, ADRs continue to account for ~10% unplanned hospital admissions in the UK. Current methods of ADR identification and management could be improved by multidisciplinary collaboration involving nurses. The aim of this study is to examine the impact of implementing the nurse-led Adverse Drug Reaction (ADRe) Profile in UK primary care. (2) Design: A pragmatic cluster-randomised controlled trial (RCT) followed by qualitative interviews in a sequential mixed-methods study. (3) Methods: For the cluster RCT, 60 patients aged ≥65 prescribed ≥5 regular medicines for long-term conditions will be recruited, 10 in each of 6 general practices. The intervention arm (*n* = 30) will complete the ADRe Profile, whilst the control participants (*n* = 30) continue to receive usual, standard care. Primary outcomes will include clinical impact on patients, benefit and prescription changes. On completion of the RCT, participants will be invited to semi-structured qualitative interviews, to evaluate the impact of the ADRe Profile from stakeholders’ perspectives, and to describe the contextual factors relevant to ADRe implementation. (4) Results: The findings of this study will evaluate the effectiveness of the ADRe Profile in identifying and resolving potential ADRs in primary care. Trial registration: This study was registered in ClinicalTrials.gov, registration number NCT04663360, date of registration—29 November 2021 (date of initial registration: 26 November 2020), protocol version 2, dated 8 January 2021.

## 1. Introduction

Medicines are prescribed for their therapeutic, preventative or diagnostic action; however, some of their complex effects on the human body and mind may be undesirable. Adverse drug reactions (ADRs) may be defined as ‘appreciably harmful or unpleasant reactions, resulting from an intervention related to the use of a medicinal product; adverse effects usually predict hazard from future administration and warrant prevention, or specific treatment, or alteration of the dosage regimen, or withdrawal of the product’ [1]. ADRs are a global problem and can cause serious harm or even death, as well as unplanned hospital admissions and raised health care costs [2,3,4,5].

It is becoming increasingly common for older people to live with multimorbidity [6,7,8] and accumulate multiple drug prescriptions, which makes them particularly susceptible to ADRs [9]. The risk of ADRs increases due to age-related physiological changes affecting pharmacokinetics and pharmacodynamics [10,11], as well as possible drug–drug interactions [12,13]. The incidence of ADRs in older hospitalised patients with dementia may be as high as 85% [14], but less data are available from UK primary care and general practices, where most medicines are prescribed and renewed [15].

Patient safety initiatives to reduce preventable medicines-related harm are high on the agenda of health organisations across the world [8,16]. In the UK, The National Institute for Clinical Excellence [4], The Kings Fund [17], and The Royal Pharmaceutical Society [18] offer guidance for medicine optimisation and management of polypharmacy. The Nursing and Midwifery Council [19] recently updated their expectations of nurses to be able to recognise and respond to ADRs.

Existing ADR-related interventions are aimed at preventing, identifying, managing or encouraging reporting of ADRs. The approach to achieving these aims may range from improving prescribing and administration of medicines, to monitoring the effects of medicines on patients’ outcomes and experiences [8,20]. However, there are no straightforward solutions without limitations. Notably, the effectiveness of comprehensive medication reviews of patient prescriptions on reducing medication-related harm in older people appeared inconclusive in a Cochrane review [21].

Meanwhile, international data indicate that preventable medicine-related harm in primary care is commonplace, affecting up to 51% of older adults following discharge from hospital [5]. Drug-related problems that actually or potentially interfere with the drugs’ intended health outcomes are experienced by 70% of older people in primary care [22].

Effective prevention, identification, and management of ADRs require clinical pharmacy expertise, as well as comprehensive monitoring [23]. Such demands require teamwork, and while the importance of interprofessional collaboration is widely acknowledged, most commonly it is the doctor–pharmacist partnership that is described in the literature [24]. The nursing contribution to prevention and management of ADRs has traditionally focused on preventing medication administration errors [25,26,27], with nurse monitoring for adverse effects of medicines often being an aspiration, rather than reality [27,28].

Nurses’ ability to detect adverse drug reactions may be limited by lack of clarity regarding their role [29] or the extent of their pharmacological knowledge [27,30], and a comprehensive tool that does not require extensive specialist expertise could facilitate such monitoring. The ADRe Profile [31] is a structured nurse-led intervention designed to help health care professionals detect adverse side effects and adverse events related to patients’ primary care medicines and to review general health and well-being, in case this has worsened as a result of prescribed medicines. It brings together relevant information and gives the prescriber or pharmacist reviewer a picture of all possible adverse effects and all medicines prescribed. It also gives users information (when clicking a link) on reporting, and on possible causes of problems identified.

Studies from community mental health settings [28,32] and care homes [3,33,34,35] demonstrate the effectiveness of the ADRe Profile in identifying and resolving health problems that may be attributed to ADRs, such as abnormal movements, postural hypotension, balance problems, falls, cognitive decline, or irritability. The scale of problems addressed ranged from bothersome (e.g., pain and sedation) to serious (seizures and dyspnoea) to potentially life threatening, such as cardiac arrhythmia with chest pain and breathlessness, and valproate-induced pancreatitis [28,32]. This illustrates how early ADR identification contributes to early resolution and medicines optimisation and may prevent more serious adverse events or deterioration.

To address the apparent lack of nursing strategies for monitoring ADRs, this study will evaluate the effectiveness of the ADRe Profile in older people in primary care receiving ≥ 5 regular medicines for long-term conditions, and gain insight into how it works and how it may be implemented in UK general practice (GP) surgeries.

## 2. Materials and Methods

This study uses an intervention developed previously with nurses [3,28,33,36,37], and was adapted for use in this project. Validity and reliability testing and a feasibility assessment were carried out, and the results will be reported separately. The evaluation stage will be guided by the 2021 Medical Research Council’s (MRC) framework for development and evaluation of complex interventions [38].

This protocol will follow the Spirit protocol checklist [39], see Appendix A.

### Design

This study has two consecutive phases: (1) a cluster-randomised controlled trial and (2) stakeholder interviews. The findings will be triangulated [40], see Figure 1 for an example. Mixed methods enhance the evaluation of a complex intervention by allowing for elements such as process evaluation to be added to the evaluation of the effectiveness [38], and complement quantitative data with insights from stakeholder accounts [41]. Survey data and qualitative data will be integrated with the outcome measures documented on the ADRe Profile, taking a pragmatic perspective of complementary triangulation [40]. Cross-cutting themes derived from the data should enhance understanding and validity, as well as contextualise the ADRe outcomes data [42,43].

## 3. Phase One: Cluster-Randomised Controlled Trial

This is a pragmatic, cluster-randomised controlled trial (RCT), following CONSORT guidelines [44] (see Appendix A). Patients in the intervention arm will complete ADRe twice. The first ADRe will be reviewed by the patients’ own primary care teams. ADRe will then be completed 1–2 months later to note any changes in reported problems or changes in care. The clinical records of all participants will be used to ascertain current problems and prescriptions at the same time as ADRe completions.

### 3.1. Participants

The clusters will be individual primary care general practitioner (GP) practices. Two sets of eligibility criteria should be considered for cluster-randomised trials [44]. At the practice level, any GP practice willing and able to participate within the pre-defined geographical area of the three local University Health Boards agreeing to participate will be eligible, and six practices will be recruited. At the participant level, members of the patients’ usual clinical teams will identify eligible patients and invite participation. The patients’ eligibility will be ascertained by screening their records.

Inclusion criteria for patient participants:Age ≥ 65 years;At least one long-term medicated condition;Prescribed ≥ 5 medications daily (vitamin and nutritional supplements and moisturising skin preparations will not be counted as ‘medicines’ for the purpose of this trial);Willing and able to give informed, signed consent themselves;Patients who, in the opinion of their nurses, lack capacity will be included if a consultee/representative is available and willing to confer with the patient and give consent on their behalf—a consultee may be a relative or friend who cares for the individual lacking capacity, but not professionally/for payment [45].

Exclusion criteria:Unable to consent and no consultee/representative available;Not fluent in English or Welsh (unless a family member can assist with translation)Receiving end-of-life care—the patient safety criteria and goals of care for patients at the end of life may be different to those of general populations [46] and specialist skills are needed to address these different challenges;Not expected to remain in the practice for the next 12 months

### 3.2. Intervention

All participants will receive usual standard care, delivered by their health care team. In addition to usual care, the ADRe Profile [3,28,47] will be implemented for the intervention arm participants (see Figure 2 for an illustrative snapshot of a part of the instrument). The control arm participants will receive only usual care but will have their medical notes and the list of prescribed medications checked for any relevant data that could be used to fill in the ADRe Profile. In both arms, any findings will be passed to a pharmacist or a prescriber, to allow identification of differences between the intervention and comparator arms.

Patients and health care professionals involved with implementing or reviewing the ADRe Profile will receive an education package on the ADRe Profile adapted from earlier studies, consisting of explanation, a booklet, frequently asked questions, and a demonstration of completing a Profile. This training will be delivered by the researcher in each of the three intervention study centres.

The ADRe Profile is described in detail elsewhere [3,47], and summarised here. The logic model underpinning the intervention is available in Appendix A. The Adverse Drug Reaction (ADRe) Profile [31] addresses the problem of unmonitored avoidable medicines-related harm by structuring communication between patients and prescribers. Nurses or carers work with patients to check the signs and symptoms that may be caused by prescription medicines, before sharing with pharmacists and prescribers; uniquely, supporting information links signs and symptoms to known effects of medicines and diseases. This ensures that reviewers have full, current patient information.

In this study, patients will be asked to complete as much of ADRe as they can themselves before meeting with their nurses. Nurses or their assistants will complete the vital signs section of ADRe with the patients.

### 3.3. Outcomes

The ADRe Profile is designed to monitor the full range of patients’ problems, including those (individually or in combinations) potentially leading to increased morbidity and mortality. These indicators are not easily detected by outcome measures such as (re)-hospitalisation rates, and there is no single measure that would adequately reflect the impact of ADRs. The effects of ADRe Profile implementation will be evaluated at individual participant level. The primary objective of this study is to explore whether the ADRe Profile identifies health problems that could be ameliorated, and record any changes to care, benefits or harms to the patients. The problems identified will be represented by the ADRe item responses that signify a clinical problem, undesirable state or suboptimal use of medicines. The problems addressed will be defined as problems where any action was taken following the problem identification (for example, medicine changes, health promotion, referral to a specialist, or other members of the multidisciplinary team). The associated outcome measures have been categorised into effectiveness and process and costs evaluation categories, but it is acknowledged that there is an unavoidable overlap:Clinical impact on patientsNew problems identified not recorded in GP notes (number and nature);Problems addressed (number and nature);Number of patients with a change in signs and symptoms potentially related to prescribed medicines (calculated as a difference between first and second ADRe Profile responses for each patient).Outcomes related to understanding the process of ADR management in primary care:
Number and nature of items on the ADRe Profile that can be populated from accessing the GP nursing and medical notes;Prescription changes (number of patients with changes in prescription regimens: drug or dose, number and nature of changes);Description of stakeholder views on ADRe Profile implementation effectiveness (survey rating of the ADRe Profile-Likert scale);Description of stakeholder views on ADRe Profile implementation feasibility (eliciting interview themes).The secondary objective is to estimate costs associated with ADRe implementation, and the associated secondary outcome measures are:
Survey of the average nurses’, assistants’, doctors’ and pharmacists’ time to complete and/or action one ADRe Profile, as mean and median length of health professionals’ time spent with one ADRe Profile.Estimation of the cost of nurses’, GP’s and pharmacists’ time based on average national salary cost per hour [48].Description of the main stakeholders’ views on multidisciplinary collaboration (eliciting interview themes).Description of the patients’ views on the contribution of ADRe Profile to patient-centred care (eliciting interview themes).

A model of the major concepts in the research framework is available in Appendix A.

### 3.4. List of Variables

Reporting all the variables potentially contributing to development of the signs and symptoms of ADRs or suboptimal medicine use is not possible in a single study due to complexity. The variables considered most relevant to the older population prescribed multiple medicines are listed:Age;Sex as m/f;Number, doses and formulations of medicines (prescribed and bought over the counter);High doses of any medicine (maximum recommended therapeutic dose);Morbidities, as reported by Davies and colleagues [49];Post code of GP practice;Involvement of a consultee y/n.

### 3.5. Sample Size

In relation to the primary outcome of the number and nature of positive changes (clinical gains) following ADRe implementation, a sample of 60 patients in 6 clusters was considered adequate in this trial. Previously, in a ‘before and after’ single-arm intervention study, with participants acting as their own controls, 17/19 (89%) had improvement in at least one clinical problem when ADRe Profiles for polypharmacy were used [3]. We hypothesised that 9/19 (47%) would have shown some improvement without intervention, necessitating an effective sample of 40 patients (20 in each arm), with 5% significance and 90% power [50]. The data provided by a sample in a cluster-randomised trial are usually thought to provide less information than if the sample were randomised by individual, due to the tendency of individuals within a cluster to share more similarities than individuals from different clusters [51]. Based on the reported intra-cluster coefficient (0.02) [35], and a design effect of 1.18, to achieve an effective sample size of 40, an actual sample size of 48 was needed [52]. Ten patients in each of 6 clusters (GP practices) allows for 20% loss to follow up, which appears realistic in primary care, where either patient or nurse may be unavailable for follow up.

### 3.6. Assignment of Interventions

The experimental design of an RCT is characterised by manipulation, control, and randomisation [53]. Cluster randomisation will allow us to regard each GP practice as an independent care provider and reduce the risk of contamination between study arms [54].

The GP surgeries will be randomised using random number-generating software (SPSS [55]) and they will be equally assigned to intervention or control arms. To avoid possible allocation bias, randomisation will be undertaken by an independent statistician in the Swansea Trials Unit, who will have no information on the patients and practices.

GP practices will be selected, and patients consented before randomisation, to reduce selection bias and to allow allocation concealment at round 1 data collection [56]. While masking of participants is not possible due to the design of this study, ascertainment bias will be lowered by researchers not being aware of the randomisation results until after having accessed the nursing and medical notes and elicited the necessary data at round 1.

### 3.7. Data Collection, Management and Analysis

The ADRe intervention will be provided to one participant at a time and will consist of the patient self-completion of the ADRe Profile, which will then be shared with the practice nurse, who will complete the vital signs and enquire about any incomplete items during a clinic appointment. Data will be collected twice for each participant, 1–2 months apart. A table of research activities is available in Appendix A. The degree to which the intervention is implemented as intended will be evaluated during the entire trial period using the modified Carroll and colleagues’ conceptual framework for implementation fidelity [57,58]. Details of the intervention core components and permitted modifications may be found in Appendix A.

### 3.8. Data Management

ADRe reports for each participating patient will be passed to the collaborating practice pharmacist, who will not be part of the research team. Allocation blinding will not be possible at this point, due to the nature of the intervention (only patients in the intervention arm will complete ADRe). Any escalation of problems to the GP or other services will be blinded. Possible causes of patients’ signs and symptoms will be ascribed, using clinical information from clinical records, including, but not restricted to, supporting information on ADRe, and the list of medicines prescribed (and bought over the counter). Juxtaposition of ADRe and medication lists is expected to generate suggestions for changes to improve the processes and possibly the outcomes of care, by relieving symptoms.

### 3.9. Statistical Methods

Data will be entered into Microsoft Excel via the electronic version of ADRe, imported into SPSS version 28 [55] for statistical analysis, checked for accuracy, meaningfulness and any missing data. The frequency of the signs and symptoms listed on ADRe will be described: as recorded in clinical notes at both stages of this study in both arms; as listed on ADRe in the intervention arm at the first and second ADRe administrations. Frequency distribution of the number of problems will be reported. Measures of central tendency will be calculated (median, mean), as well as variability (the range, standard deviation, and quartiles).

Baseline characteristics of participating GP surgeries and patients will be analysed to assess cluster differences between the intervention and control arms. Potentially confounding variables (age, number of prescribed medicines, and number of co-morbidities) will be identified and described, with a view to incorporating them in the statistical analysis. Differences between pre- and post-intervention values will be calculated for: problems present (all ADRe items separately and together), problems addressed (all reported separately and together), and compared for all participants and by site. The magnitude of effect or effect size will be calculated.

To account for cluster, and patient factors, such as age and number of prescribed medicines, a logistic regression model will be built, entering predictor variables individually. The impact of age, diagnoses, and medicines’ use (numbers and types of medicines prescribed) will also be considered.

The secondary outcomes of resource use and costs will be described using the Personal Social Services Research Unit [48] unit costs of health workers and the National Tariff Payment Systems [59]. Estimated quarterly administration costs will be offset against quantified burden of adverse drug events in primary care in the UK [2].

### 3.10. Data Monitoring

Since this clinical trial does not involve investigation of medicinal products, a data safety and monitoring board is not required. Our trial steering group will review the data and respond to researchers’ concerns every 3 months.

### 3.11. Criterion for Stopping This Study Early

If any subject experiences a serious adverse event (defined in the protocol) possibly due, in the opinion of researchers or the study pharmacists, to administration of ADRe, research activities for all participants will be suspended until the steering committee can convene to review evidence for causality. For further information on safety measure plans, please see Appendix A.

### 3.12. Governance

An independent steering committee has been formed to oversee adherence to key performance indicators and setting goals for improvement and measures. The steering committee will meet a further 4 times throughout the duration of the trial and comprises a consultant pharmacist, a primary care pharmacist and 2 patients. This study is being overseen by Swansea University as sponsor. Standard Operating Procedures of Swansea Clinical Trials Unit are being used, with permission.

### 3.13. Ancillary and Post-Trial Care

Participants in the control arm will be offered the ADRe Profile once this study finishes, if interested. All participants will be invited to register to use the ADRe Profile, without cost.

## 4. Phase Two: Stakeholder Views

The last phase of the research project will be a stakeholder survey, comprising structured questionnaires and in-depth interviews, further supplemented by field note data that will be collected during the RCT. The purpose of this phase is to obtain the end users’ evaluation of ADRe, plus views on barriers and enablers of its implementation, acceptability and suitability. This study builds on professionals’ and participants’ perspectives and understanding of the functioning of ADRe captured in earlier evaluations of the process involved in ADRe administration [3,28,35]. This phase will widen the effectiveness focus to include a broader consideration of core elements that will help the understanding of specific reasons causing the intervention to succeed or fail through process evaluation [38]. The participants will be the health professionals and patients from the intervention arm.

A structured questionnaire with 3-item Likert-style response options will document all participants’ views based on their experience of involvement with the ADRe Profile. This will capture stakeholders’ views on ADRe’s effectiveness and the feasibility of its integration in general practice. The questions build on earlier work [35] and relevant patient-centred care quality indicators, as identified by Santana and colleagues [60]. For the questions and response options, see Appendix A.

The semi-structured, in-depth interviews will be analysed using an inductive approach involving thematic analysis [61]. This involves a basic form of thematic analysis combined with use of the constant comparative method [62,63], rather than the complex, nested coding associated with competing versions of grounded theory [64]. Grounded theory is widely used in qualitative interview studies [65] and will facilitate developing ideas generated in early interviews as the interview work progresses to generate credibility, originality, resonance and usefulness [66]. The comparative aspect implies that “emerging codes, categories, properties, and dimensions as well as different parts of the data, are constantly compared with all other parts of the data to explore variations, similarities and differences in data” [61] (p. 143). The interviewer will use an interview guide to adjust questions, which, while covering the main topics associated with the study objectives [67], will allow respondents to describe their experiences and opinions in their own words, and communicate relevant insights that were not anticipated when the guide questions were formulated [68].

The aim is to gain a well-rounded insight into stakeholder perspectives on the intervention (and its implementation), which will supplement and illustrate the RCT data. A purposive sample of 12 interviewees will be selected from a pool of participants who have completed the structured questionnaire. The questionnaire results will enable maximum variation sampling [69]. The interviews will be conducted by telephone, videoconference or in person, accommodating interviewees’ preferences. The interview guide will focus data collection and ensure coverage of the key aspects of interest. The template guide (Appendix A) will support logical flow of topics/questions, yet at the same time, the inherent flexibility of semi-structured interviews will allow deviations depending on how the interviewee responds to the questions. Data collection and analysis will proceed concurrently, reflecting the constant comparative method [60]. Data from the interviews and field notes will be coded, categorised and conceptualised by 2 researchers.

## 5. Ethics and Dissemination

The regulatory and governance requirements were met through approval from the relevant review bodies. The ethical approval for this study was granted by HRA/HCRW (IRAS ID: 292693, date: 10 March 2021) and the institutional Research Ethics Committee (ref: 080321b, date: 9 June 2021).

### 5.1. Consent

Written consent will be sought from all phase one and phase two participants. For people who lack capacity, a personal consultee, who has an unpaid or non-professional role in caring for the person (often a close family member), will be asked for advice [45]. If the consultee advises that the participant would have wanted to take part in the research had they not lacked capacity, and if the consultee is willing to assist the participant with their involvement in this study, then the participant will be recruited.

### 5.2. Confidentiality

Participant data will be processed in accordance with the Data Protection Act 2018 [70]. In the trial, the records of the ADRe Profile will be viewed only by the research team and the participants’ usual nurse, pharmacist and GP. All participants will be allocated a study number to ensure anonymity during data analysis and publication of results. No identifiable information will be collected during the interviews.

### 5.3. Dissemination

On completion of this study, an executive summary of the research findings will be freely available in participating general practices or by contacting the researchers. The findings will be presented at seminars and conferences and published in open-access academic papers.

## 6. Discussion

This pragmatic cluster RCT is innovative in its approach to the issue of ADR monitoring. Firstly, the nurse-led intervention assists strong multi-professional collaboration in ensuring patient safety in primary care by building on clinical partnership between the nurse, pharmacist and prescriber or GP. Secondly, the intervention that is being tested enables nurses’ involvement in ADR monitoring and resolution of any problems by its incorporation of comprehensive supporting information and clear signposting of suggested actions. Thirdly, the ADRe Profile supports patient-centred care by its inclusion of patient-valued outcomes. This study may have implications for medicine optimisation in primary care by contributing to ADR monitoring, prevention and management.

This study has several limitations, pertaining to both phases of this study and the intervention itself. ADRe appears long. It was designed to be comprehensive [28,36,37], to offer practitioners a single document for communication, to obviate the need for pharmacists and doctors to review the numerous (up to 40) documents seen in many online systems. When evaluating ADRe, clinicians have indicated that all items are important and reflect reality [3]. Although ADRe aims to manage most potential ADRs, it is possible that the services will not have sufficient time to address the numerous issues that may arise, many of which will have multiple aetiologies, including multiple medicines and co-morbidities. The patient population in the sample of six participating general practices may not be representative of the wider UK population and volunteer bias may affect this study both at the practice and the patient participant level [68]. This will have implications for the generalisability of the study findings. An important RCT limitation is that blinding is not achievable in this study and both participants and personnel will be aware of the assigned study arm. This may lead to bias in the measurement of the outcome [69,70], where both participant-reported outcomes and experimenter expectations are influenced by participant’s knowledge of their assigned arm, limiting the strength of any causal inferences. The validity of the statistical analysis may also be affected by the self-reported nature of some ADRe items, their inherent subjectivity, and possible fluctuation [68]. It may be impossible to separate the signs and symptoms of morbidities from ADRs, as the signs and symptoms may be the same; however, these signs and symptoms need to be addressed, regardless of aetiology. The data obtained in the interviews may not be generalisable to wider population of patients and health care staff using the ADRe Profile [68]. Finally, the costs obtained for implementation are not based on a full economic model: rather, they will estimate the balance between professionals’ time needed and benefits to patients, based on costings derived from the literature.

## 7. Conclusions

ADRs are persistent and often preventable problems, ranging from relatively mild (but often bothersome) to life-threatening issues. Older adults prescribed multiple medicines are particularly at risk from ADRs. There are very few systems for ADR monitoring for multiple medicines. This study protocol describes a clinical trial of the ADRe Profile, which identifies and helps resolve clinical problems, undesirable states or suboptimal use of medicines. The ADRe Profile brings together multiple professionals to keep patients safe and to ultimately reduce staff workload.

## Figures and Tables

**Figure 1 pharmacy-10-00052-f001:**
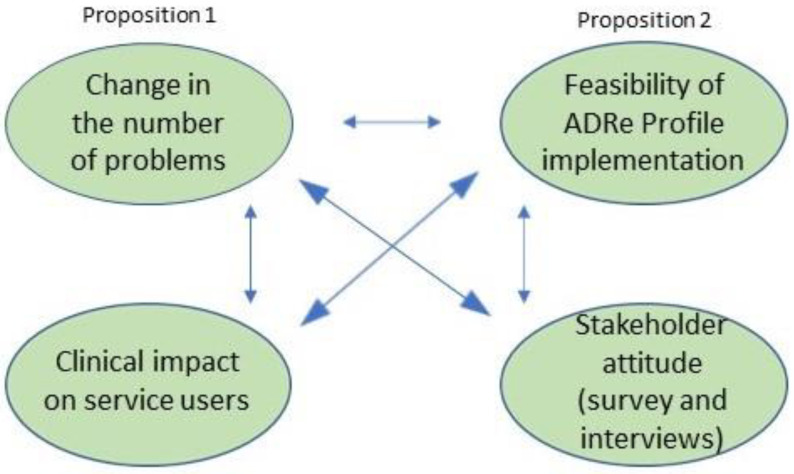
Illustrating the complementary triangulation.

**Figure 2 pharmacy-10-00052-f002:**
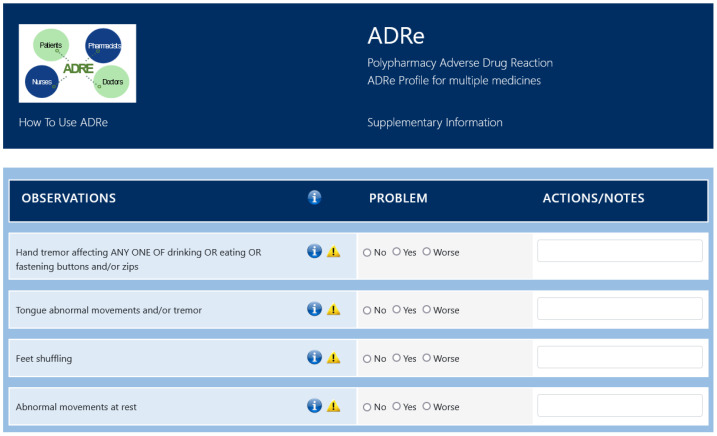
Example section of the ADRe Profile, demonstrating the supplementary information icons.

## Data Availability

The data used in this study will be available to the research data community at https://zenodo.org/record/4090384#.X4hKptZFzLZ. All proposals to view the data are subject to review by Swansea University’s Research Governance department and the PI. Before any data can be accessed, approval must be given. The application process is via the Academic Lead for Research Integrity Research Engagement & Innovation Services, Swansea University and the PI or Neil Carter. Contacts: Swansea University, Swansea SA2 8PP • Tel: +44-0-1792-606060 and 518541 or 295610 • Email: researchgovernance@swansea.ac.uk, s.e.jordan@swansea.ac.uk or n.carter@swansea.ac.uk. The research instrument used in this study is available for clinical use without charge via the project website: http://www.swansea.ac.uk/adre/.

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
