# Peer review of "Clinical Impact of Implementing a Nurse-Led Adverse Drug Reaction Profile in Older Adults Prescribed Multiple Medicines in UK Primary Care: A Study Protocol for a Cluster-Randomised Controlled Trial"

_pharmacy, 2022, doi:10.3390/pharmacy10030052_

Round 1

Reviewer 1 Report

The following comments, clarifications or changes are raised:

  • The proposed keywords should be DeCS / MeSH
  • A brief explanation should be included in the abstract about ADR identification and management.
  • Exclusion criteria for palliative care patients should be explained
  • In the statistical analysis it is important to refer to the fact that the magnitude of effect or effect size will be calculated.
  • The clinical trial must be registered in the international registry of clinical trials
  • It must be justified why the grounded theory will be used as a qualitative study design and not an ethnography that will be determined by the social and cultural characteristics of the population under study.

Author Response

Thank you for your comments,

  1. The proposed keywords should be DeCS / MeSH

MeSH Keyword

RDF Unique Identifier

drug-related side effects and adverse reactions

http://id.nlm.nih.gov/mesh/D064420

polypharmacy

http://id.nlm.nih.gov/mesh/D019338

primary health care

http://id.nlm.nih.gov/mesh/D011320

aging

http://id.nlm.nih.gov/mesh/D000375

nurses

http://id.nlm.nih.gov/mesh/D009726

randomised controlled trial

http://id.nlm.nih.gov/mesh/D016449

  1. A brief explanation should be included in the abstract about ADR identification and management.

The abstract’s word limit precludes more than a sentence. We have added:

‘The professional bodies of nursing, medicine and pharmacy have issued guidelines on identification and management of ADRs; however, these continue to account for ~10% unplanned hospital admissions in the UK.’

As indicated in the introduction, this is the subject of an earlier paper.

  1. Exclusion criteria for palliative care patients should be explained

Thank you for pointing out this issue. In section 3.1, we changed the term ‘palliative’ to ‘end of life’ (the patient safety criteria and goals of care for patients at the end of life may be different to those of general populations (Akdeniz et al., 2021) and specialist skills are needed to address these different challenges), to reflect the distinction between ‘palliative care’ as care for people who are facing the challenges associated with terminal illness (World Health Organisation [WHO], 2020) and end of life care as support and care given to people during time surrounding their death (National Institute of Ageing, 2021). The patient safety criteria and the goals of care for patients at end of life care are not necessarily those of the general population (Akdeniz et al., 2021). Specialist skills are needed to address these very different challenges and understand the therapeutic regimens prescribed by specialists. This study focuses on medicines prescribed by general practitioners.

  1. In the statistical analysis it is important to refer to the fact that the magnitude of effect or effect size will be calculated.

The following has been added in section 3.9: ‘The magnitude of effect or effect size will be calculated.’

  1. The clinical trial must be registered in the international registry of clinical trials

The following has been added to page 1: Trial registration: The study was registered in ClinicalTrials.gov, registration number NCT04663360, date of registration – 29/11/2021 (date of initial registration: 26/11/2020), protocol version 2, dated 08/01/2021.

  1. It must be justified why the grounded theory will be used as a qualitative study design and not an ethnography that will be determined by the social and cultural characteristics of the population under study.

We have opted for a grounded theory because we viewed the methodology as more suitable to fulfill research aims. Viewed as a process evaluation, the qualitative part of the study is designed to enhance the understanding of the core processes underpinning the intervention. Besides, available resources preclude long periods of sustained fieldwork effort, grounded theory is thus more realistic for a study of the impact of ADRe.

GT has been widely used in interview studies as well as ethnographies (as for example in Kathy Charmaz’s classic study Good days, Bad Days. Charmaz was a leading authority on GT.

Ethnographies general focus on a setting or cultural group whose members maintain regular contact.  Few ethnographies are concerned with samples of persons who share some characteristic but do not interact often.  Qualitative studies of samples of persons such as mothers of Downs children or older people prescribed polypharmacy are usually interview studies. This we believe that the reviewer is mistaken in implying that the existence of common social and cultural characteristics alone is sufficient to create the conditions necessary for a participant observation study.

The following text has been added to section 4: ‘The grounded theory is an approach widely used in qualitative interview studies (Charmaz & Belgrave, 2012) and has been chosen as most suitable to fulfil the research aims.’

Literature:

Akdeniz, Yardımcı, B., & Kavukcu, E. (2021). Ethical considerations at the end-of-life care. SAGE Open Medicine, 9, 20503121211000918–20503121211000918. https://doi.org/10.1177/20503121211000918

Charmaz, & Belgrave, L. L. (2012). Qualitative Interviewing and Grounded Theory Anal-ysis. In The SAGE Handbook of Interview Research: The Complexity of the Craft (2nd ed., pp. 347–366). SAGE Publications, Inc. https://doi.org/10.4135/9781452218403.n25

National Institute of Ageing. (2021). Providing Care and Comfort at the End of Life. Accessed at: https://www.nia.nih.gov/health/providing-comfort-end-life on 06/04/2022.

World Health Organisation (WHO). (2020). Palliative care. Accessed at https://www.who.int/news-room/fact-sheets/detail/palliative-care on 06/04/2022.

Reviewer 2 Report

Overall, I enjoyed reading this study. As a pharmacist practicing in the U.S., it was interesting to see a nurse-led ADR profile system in the UK. Since it is a protocol, there is no result to be reported. Still, after carefully reading the article and supplementary document, I found the design of the study appropriate, as well as following good clinical practice. The only question I had was a relatively small sample size (60 patients total), which I feel can be increased to 100 or more. Other than that, as long as the authors pay more attention to grammar and punctuation errors and address some of the comments I left in the reviewed manuscript, this paper will be good to go. Thank you so much for your hard work and I wish you the best of luck in conducting the actual research.

Author Response

Thank you.

We agree that a larger study would be desirable. However, a sample of 60 patients will be sufficient to ascertain whether the benefits observed in care homes (Jordan et al 2015, 2019, 2021), respiratory care (Gabe et al 2014) and mental health settings (Jones et al 2016, Jordan et al 2002) transfer to primary care.

Literature:

Gabe, Marie E, Murphy, Fiona, Davies, Gwyneth A, Russell, Ian T, & Jordan, Susan. (2014). Medication monitoring in a nurse-led respiratory outpatient clinic: pragmatic randomised trial of the West Wales Adverse Drug Reaction Profile. PloS One, 9(5), e96682–e96682. https://doi.org/10.1371/journal.pone.0096682

Jones, Richard, Moyle, Christopher, & Jordan, Sue. (2016). Nurse-led medicines monitor-ing: a study examining the effects of the West Wales Adverse Drug Reaction Profile. Nursing Standard, 31(14), 42–53. https://doi.org/10.7748/ns.2016.e10447

Jordan. (2002). Managing adverse drug reactions: an orphan task. Journal of Advanced Nursing, 38(5), 437–448. https://doi.org/10.1046/j.1365-2648.2002.02205.x

Jordan, S., Gabe, M., Newson, L., Snelgrove, S., Panes, G., Picek, A., Russell, I.T., & Dennis, M. (2014). Medication Monitoring for People with Dementia in Care Homes: The Feasibility and Clinical Impact of Nurse-Led Monitoring. TheScientificWorld, 2014, 843621–11. https://doi.org/10.1155/2014/843621

Jordan, S., Gabe-Walters, M.E., Watkins, A., Humphreys, I., Newson, L., Snelgrove, S., & Dennis, M.S. (2015). Nurse-Led Medicines' Monitoring for Patients with Dementia in Care Homes: A Pragmatic Cohort Stepped Wedge Cluster Randomised Trial. PloS One, 10(10), e0140203–e0140203. https://doi.org/10.1371/journal.pone.0140203

Jordan, S., Banner, T., Gabe-Walters, M., Mikhail, J.M., Panes, G., Round, J., Snelgrove, S., Storey, M., & Hughes, D. (2019). Nurse-led medicines’ monitoring in care homes, im-plementing the Adverse Drug Reaction (ADRe) Profile improvement initiative for mental health medicines: An observational and interview study. PloS One, 14(9), e0220885–e0220885. https://doi.org/10.1371/journal.pone.0220885

Jordan, Prout, H., Carter, N., Dicomidis, J., Hayes, J., Round, J., & Carson-Stevens, A. (2021). Nobody ever questions-Polypharmacy in care homes: A mixed methods evaluation of a multidisciplinary medicines optimisation initiative. PloS One, 16(1), e0244519–e0244519. https://doi.org/10.1371/journal.pone.0244519

Round 2

Reviewer 1 Report

After evaluating the response of the authors to the comments raised, the changes are considered to have improved the clarity of the manuscript.